# The Crosstalk and Clinical Implications of CircRNAs and Glucose Metabolism in Gastrointestinal Cancers

**DOI:** 10.3390/cancers15082229

**Published:** 2023-04-10

**Authors:** Xiaonuan Luo, Yin Peng, Xinmin Fan, Xiaoxun Xie, Zhe Jin, Xiaojing Zhang

**Affiliations:** 1Guangdong Provincial Key Laboratory of Genome Stability and Disease Prevention and Regional Immunity and Diseases, Department of Pathology, Basic Medicine School, Shenzhen University Medical School, Shenzhen University, Shenzhen 518060, China; xiaonuanluo@126.com (X.L.);; 2Department of Histology and Embryology, School of Pre-Clinical Medicine, Guangxi Medical University, Nanning 530021, China

**Keywords:** circular RNAs, glucose metabolism, gastrointestinal cancers, biomarkers, therapeutic targets

## Abstract

**Simple Summary:**

Glucose metabolism reprogramming is a hallmark of cancer. In tumor cells, the majority of glucose is converted to lactate despite the presence of sufficient oxygen and functional mitochondria, thereby contributing to cancer progression and immunosuppression. Circular RNAs are a type of endogenous single-stranded RNAs characterized by covalently circular structures. Recent studies have revealed that circular RNAs regulate glucose metabolism in various cancers. In this review, we discuss the molecular mechanisms by which circular RNAs regulate glucose metabolism in gastrointestinal cancers. Furthermore, we consider the potential of glucose-metabolism-associated circular RNAs as new biomarkers and therapeutic targets for gastrointestinal cancers.

**Abstract:**

The majority of glucose in tumor cells is converted to lactate despite the presence of sufficient oxygen and functional mitochondria, a phenomenon known as the “Warburg effect” or “aerobic glycolysis”. Aerobic glycolysis supplies large amounts of ATP, raw material for macromolecule synthesis, and also lactate, thereby contributing to cancer progression and immunosuppression. Increased aerobic glycolysis has been identified as a key hallmark of cancer. Circular RNAs (circRNAs) are a type of endogenous single-stranded RNAs characterized by covalently circular structures. Accumulating evidence suggests that circRNAs influence the glycolytic phenotype of various cancers. In gastrointestinal (GI) cancers, circRNAs are related to glucose metabolism by regulating specific glycolysis-associated enzymes and transporters as well as some pivotal signaling pathways. Here, we provide a comprehensive review of glucose-metabolism-associated circRNAs in GI cancers. Furthermore, we also discuss the potential clinical prospects of glycolysis-associated circRNAs as diagnostic and prognostic biomarkers and therapeutic targets in GI cancers.

## 1. Introduction

Gastrointestinal (GI) cancers include gastric cancer (GC), hepatocellular carcinoma (HCC), colorectal cancer (CRC), esophageal cancer (EC) and pancreatic cancer (PC) [1]. GI cancers are among the most prevalent tumors worldwide, accounting for approximately 35.4% of all cancer-related deaths and about 26.3% of the global cancer incidence [2]. The primary treatment for GI cancers is the surgical resection of tumor; however, distant metastases arise in many patients, leading to death. Therefore, it is necessary to elucidate the molecular mechanisms underlying the occurrence and metastasis of GI cancers and identify new biomarkers and therapeutic targets to improve outcomes in patients with GI cancers.

Glucose is an essential source of energy and carbon for various biochemical reactions as well as cell proliferation [3]. Glucose is initially converted to pyruvate in the cytoplasm. In quiescent cells under normoxic conditions, pyruvate is metabolized via oxidative decarboxylation and then enters the tricarboxylic acid (TCA) cycle. The two processes produce NADH and FADH2 molecules that further undergo oxidative phosphorylation (OXPHOS) in the mitochondrial inner membrane [4], thereby generating adenosine triphosphate (ATP) as an energy source [5]. Alternatively, under hypoxic conditions, pyruvate is reduced to lactate in the cytoplasm in a process known as anaerobic glycolysis [6]. In 1924, Otto Warburg observed that the majority of glucose is converted to lactate in cancer cells, even in the presence of abundant oxygen, a phenomenon known as the “Warburg effect” or “aerobic glycolysis” [7]. This reprogramming of glucose metabolism has now been identified as a key hallmark of cancer [8]. 

Noncoding RNAs, including microRNAs (miRNAs), long non-coding RNA (lncRNAs), and circular RNAs (circRNAs), as well as transcription factors are involved in regulating glucose metabolism by controlling the transcription, protein stability, and enzymatic activity of the key enzymes and transporters of glycolysis in cancer cells [9,10]. CircRNAs were first observed in viroids by Sanger et al. in 1976. At that time, circRNAs were considered to be “junk” RNA generated through the abnormal splicing of precursor mRNA (pre-mRNA) during post-transcriptional modification processes [11]. In recent years, major advances in high-throughput RNA-sequencing and bioinformatics techniques have revealed the vast diversity of circRNAs, which have been identified in a variety of eukaryotes, including fish, fungi, and mammals. Moreover, circRNAs have been shown to play vital roles in both physiological and pathological processes [12]. Accumulating evidence shows that circRNAs are widely dysregulated in various cancers, and play key roles in tumor cell proliferation, metastasis and drug resistance [13].

Here, we provide a comprehensive review of the relationship between circRNAs and glucose metabolism in GI cancers. We summarize the circRNAs that have been shown to influence glucose metabolism by regulating the specific glycolytic enzymes or transporters. We also discuss the mechanisms by which circRNAs control the glycolytic phenotype through pivotal signaling pathways (Table 1). In addition, we highlight the potential of circRNAs as biomarkers and therapeutic targets for clinical applications.

## 2. Glycolysis Contributes to GI Cancers Progression

Increased glycolysis, including increased glucose uptake, as well as abundant lactate and ATP production, has been documented in EC, GC, CRC, HCC, and PC [14,15,16,17,18]. Many glucose-metabolism-related transporters and enzymes are upregulated and promote glycolysis in GI cancer. Glucose transporter 1 (GLUT1) is the primary transporter of glucose and mediates glucose uptake as the first step in glycolysis. GLUT1 was found to be highly expressed in GC and EC, which exhibit increased glycolysis [19,20]. In addition, the three rate-limiting enzymes in the glycolytic pathway, i.e., hexokinase 2 (HK2), phosphofructokinase (PFK), and pyruvate kinase (PK), are involved in reprogramming of glucose metabolism in GI cancers. HK2 in CRC, PC, and HCC [21,22,23] as well as PFK in CRC [24], and PKM2 in GC and PC [25,26], have been reported to be upregulated and to enhance glycolysis. 

A schematic diagram of the mechanism by which glycolysis contributes to GI cancer progression is shown in Figure 1. Glycolysis produces large amounts of ATP and NADH, as well as the raw materials for the biosynthesis of macromolecules required to support cell proliferation. ATP is a universal biochemical component of glucose metabolism in all cells [27,28]. Glycolysis generates only two ATP molecules per glucose molecule, whereas oxidative phosphorylation produces up to 36 ATPs per glucose molecule [16]. In cancer cells, a large proportion of the available pyruvate is reduced into lactate and only a few pyruvate molecules enter the TCA cycle for oxidative phosphorylation [5,6,29]. This raises the questions of why tumor cells adopt the glycolytic phenotype and how the requirement for abundant ATP to support uncontrolled cell proliferation is met. Studies of various cancers have indicated that cancer cells absorb 10–100 times more glucose than surrounding normal cells, thereby producing sufficient ATP and providing a growth advantage for cancer cells [30]. 

Glycolysis contributes to the synthesis of macromolecules [6]. The accumulated glycolytic intermediates supply raw materials for the biosynthesis of nucleotides, lipids and proteins to generate new cells [8]. The glycolytic intermediate glucose-6-phosphate (G6P) is converted to ribose-5-phosphate via the pentose phosphate pathway (PPP), which supplies the raw materials for the synthesis of ribonucleotides [31]. Furthermore, the glycolytic intermediates dihydroxyacetone phosphate and 3-phosphoglycerate are required for the biosynthesis of lipids and amino acids, respectively [32]. In addition, G6P is also involved in the production of nicotinamide adenine dinucleotide phosphate oxidase (NADPH) via the PPP to protect cancer cells against reactive oxygen species (ROS) [33]. 

## 3. Glycolysis Induces an Immunosuppressive Tumor Microenvironment in GI Cancers

In addition to promoting cancer cell proliferation, glycolysis is also involved in immune escape. Initially, the tumor microenvironment (TME) comprises not only cancer cells, but also non-cancerous host cells, including innate and adaptive immune cells [34]. The glycolytic phenotype of cancer cells inhibits the immune defense function of these immune cells by competitively consuming glucose and restricting the availability of glucose for immune cells [35]. As the disease progresses, cancer cells generate abundant lactate as the end-product of glycolysis [36]. The lactate is then secreted into the TME via the monocarboxylate transporter 1 (MCT1). Although lactate was initially regarded as a waste-product of glucose metabolism, accumulating evidence suggests that lactate can induce immunosuppression in GI cancers by modulating the inflammatory microenvironment. As shown in Figure 1, lactate induces the differentiation of M2 phenotype macrophage. Macrophages are the dominant immune cell type in the TME. Based on their polarized state, macrophages are categorized as M1 and M2 phenotypes. While M1 macrophages inhibit cancer progression, M2 macrophages are involved in the anti-inflammatory responses that induce immunosuppression and promote cancer progression. Lactate in the TME enters macrophages and increases HIF1α expression, which promotes the conversion of M1 macrophages to the M2 phenotype by upregulating the M2 marker ARG1 [37]. Furthermore, lactate enhances the function of regulatory T cells. Regulatory T cells, also known as suppressive T cells, were initially discovered in the early 1970s by Gershon et al. Chemokines in the TME induce recruitment of regulatory T cells to inhibit T cell immunity against tumor self-antigens. Thus, regulatory T cell-mediated immunosuppression is one of the key mechanisms of tumor immune escape. Lactate is involved in regulating the function of regulatory T cells via the transforming growth factor β (TGF-β) signaling pathway. In regulatory T cells, lactate contributes to MOESIN lactylation at Lys72 to enhance its interaction with TGF-β receptor I (TGF-β RI) and MOESIN. This interaction upregulates TGF-βRI-associated SMAD3-FOXP3 signaling to enhance regulatory T cell function [38].

Lactate has also been shown to promote cancer metastasis. CRC cell-derived lactate increases the expression of CXCL10 and cadherin-11 in CD115^+^ precursors via the PI3K-AKT-CREB pathway and mammalian target of rapamycin (mTOR) signaling pathways, resulting in the formation of osteolytic lesions and a metastatic niche that promotes bone metastasis of CRC [39].

## 4. The Characteristics of Glycolysis-Associated CircRNAs in GI Cancers

CircRNAs are a type of endogenous RNA generated through a back-splicing process in which the 3′ splice donor covalently ligates to the 5′ splice acceptor, forming cyclic structures and characteristic junction sites. Based on their specific composition, circRNAs are classified as intronic, exonic, and exon-intron circRNAs [13]. In GI cancers, most glycolysis-associated circRNAs are produced from exons of pre-mRNAs, which are mainly located in the cytoplasm. Numerous circRNAs have been shown to be aberrantly expressed in GI cancers and involved in cancer progression. Upregulated expression of circAXIN1 [40], and circNRIP1 [41] has been reported to enhance cell migration, proliferation, and invasion in GC, while downregulated expression of circ-HuR [42], circDIDO1 [43] and circURI1 [44] in GC has the opposite effect on GC progression. Dysregulated circRNAs may also be involved in GI cancer metastasis, a process in which lymphangiogenesis plays an active role. CircNFIB1 is expressed at low levels in patients with PC, and circNFIB1 knockdown induces lymphangiogenesis to promote lymph node metastasis of PC [45]. Furthermore, circRNAs are associated with drug resistance. In CRC, circNRIP1 [46] and circ_0000338 [47] confer 5-fluorouracil (5-FU) resistance and contribute to CRC progression, whereas circ_0082182 [48], hsa_circ_0005963 [49] and circATG4B [50] confer resistance to oxaliplatin.

CircRNAs play a key role in cell biology by acting as miRNA sponges, protein sponges or encoding novel proteins. In GI cancers, the majority of circRNAs identified so far have been proposed to modulate glucose metabolism by competitive binding or sponging miRNAs via miRNA response elements (MREs), which specifically binds and sponges miRNAs to block their ability to suppress the expression of their target genes. In this way, circRNAs can indirectly increase miRNAs-associated target gene expression [51]. For example, circMAT2B functions as a miRNA sponge to increase glycolysis in HCC. Li et al. showed that miR-338-3p downregulates the expression of its target gene in HCC by direct interaction with the 3’-UTR of the target gene, and that this effect is reversed by the sponge activity of circMTA2B [52]. Expression of circGOT1, derived from exon 8 of its host gene *GOT1*, and GOT1 are upregulated in EC tissues, contributing to glucose metabolism and disease progression. According to bioinformatic analysis with Targetscan and the Circular RNA Interactome, miR-606 is predicted to interact with circGOT1 and *GOT1*. Mechanistically, circGOT1 contributes to GOT1 expression by sponging miR-606 in EC [53]. Furthermore, circRNAs may be involved in glucose metabolism by binding two or more miRNAs. For example, circFAT1 upregulates UHRF1 to promote CRC cell proliferation and glycolysis by targeting miR-520b and miR-302c-3p [54]. Zhao et al. also showed that circATP2B1 increases PKM2-mediated glycolytic metabolism in GC by binding with miR-326-3p and miR-330-5p [55]. Although the circRNA-miRNA interaction is the classical mechanism of circRNA function, a recent computational study predicted that only a fraction of circRNAs harbor miRNA binding sites [56]. Hence, the possibility that miRNAs function as sponges of circRNAs requires verification. 

Several studies have demonstrated that many circRNAs contain binding sites for proteins, indicating that these circRNAs may function as protein sponges [57]. Evidence suggests that circRNAs bind with the target proteins to regulate their stability in GI cancers. CircACC1 binds to the regulatory β and γ subunits of AMPK to increase AMPK holoenzyme stability and activity, which promotes glycolysis and induces CRC progression [58]. CircRPN2 interacts with ENO1 to promote the degradation of ENO1 and inhibits glycolysis in HCC [59]. In addition, circRNAs may regulate the intracellular translocation of their target proteins in GI cancers. CircRHBDD1 sponges and recruits YTHDF1 to the PIK3R1 promoter to drive the translation of PIK3R1, which promotes glycolysis and inhibits anti-PD-1 therapy in PC [60] (Table 2).

## 5. CircRNAs Regulate Glycolysis-Related Enzymes or Transporters in GI Cancers

Accumulating evidence suggests that circRNAs influence cancer progression by controlling glycolysis via regulating specific glycolysis-associated enzymes and transporters in various cancers. In GI cancers, circRNAs reprogram glucose metabolism by regulating the transcription or expression of GLUT1, HK2, PKM2, and LDHA1 (Figure 2).

### 5.1. CircRNAs and GLUT1 in GI Cancers

Increased glycolysis is characterized by elevated glucose uptake. GLUT proteins encoded by *SLC2* genes are integral membrane proteins are involved in the movement of glucose across the cell membrane. To date, 14 different GLUT isoforms have been identified, each with its own unique tissue distribution. High expression of GLUT1 and GLUT3 has been identified in various cancers and these proteins have been shown to play an important role as glucose transporters in cancer cells [61].

CircRNAs can regulate GLUT1 expression to reprogram glucose metabolism. For example, circDENND4C is upregulated in CRC tissues and cells, and promotes glycolysis by inducing GLUT1 expression. Zhang et al. discovered that circDENND4C functions as a sponge for miR-760 to inhibit its activity, and also increases the expression of its target gene, GLUT1, in CRC. Thus, circDENND4C contributes to the glycolytic phenotype in CRC through the circDENND4C/miR-760/GLUT1 axis [62]. However, as yet, there have been no reports describing the role of circRNAs in regulating GLUT3 expression in GI cancers.

### 5.2. CircRNAs and HK2 in GI Cancers

Hexokinases (HKs) are evolutionarily conserved enzymes that catalyze the first step of glucose metabolism to produce G6P. In mammals, five HK isoforms have been identified: HK1–4 and hexokinase domain containing protein 1 (HKDC1). Of these, HK2 primarily exists in insulin-sensitive tissues and cancer cells and is involved in glucose metabolism [63,64].

CircBFAR, which is derived from exon 2 of the human *BFAR* gene, augments GC progression by directly sponging miR-513a-3p to increase HK2 expression and upregulate glycolysis [65]. Hsa_circ_0045932 in CRC [66], as well as hsa_circ_0001806 and circ-PRMT5 in HCC [67,68], also accelerate glycolysis by upregulating HK2 expression through their ability to sponge miR-873-5p, miR-125b and miR-188-5p, respectively. 

CircCUL3 acts as a sponge for miR-515-5p to enhance the expression of signal transducer and activator of transcription 3 (STAT3), which then binds to the HK2 promoter to activate HK2 transcription and increase glycolysis in GC [69]. 

### 5.3. CircRNAs and PKM2 in GI Cancers

Pyruvate kinase M2 (PKM2) catalyzes the conversion of phosphoenolpyruvate into pyruvate as a vital rate-limiting step in glucose metabolism [70,71]. CircMAT2B [52] and hsa_circ_0005963 [49] upregulate PKM2 expression and facilitate glycolysis and HCC progression by sponging miR-338-3p and miR-122, respectively. CircATP2B1 contains many miRNA binding sites, and can sponge both miR-326-3p and miR-330-5p, which are members of the miR-326 gene cluster, to promote PKM2 expression, thus increasing glycolysis [55].

### 5.4. CircRNAs and LDHA in GI Cancers

Lactate dehydrogenase A (LDHA) plays a crucial role in glycolysis by catalyzing the conversion of pyruvate into lactate [72]. CircUBE2D2 in HCC [73] and circ-DONSON in GC [74] increase LDHA expression by sponging miR-889-3p and miR-149-5p, respectively, to induce glycolysis. 

CircRNAs also affect the glycolytic phenotype of cancer cells by regulating LDHA transcription. CircPLOD2 acts as a sponge for miR-513a-5p to increase the expression of SIX1, which is then recruited to the LDHA promoter to induce its transcription and enhance glycolysis in CRC [75]. CircSLIT2 activates the c-Myc transcription factor by functioning as a sponge for miR-510-5p, to promote LDHA transcription in PC [76].

## 6. CircRNAs Control Glycolysis by Regulating Signaling Pathways

### 6.1. CircRNAs Are Involved in Glycolysis by Regulating the HIF1, c-Myc, or STAT3 Signaling Pathways in GI Cancers

A schematic diagram of the proposed mechanism by which circRNAs control glycolysis by regulating HIF1, c-Myc, or STAT3 signaling in GI cancers is shown in Figure 3.

#### 6.1.1. CircRNAs and HIF1 in GI Cancers

Dysregulated proliferation of cancer cells and insufficient supply of blood vessels limit the supply of nutrients and oxygen, creating a hypoxic microenvironment. Hypoxia inducible factor 1 (HIF1) is a heterodimeric complex that contains an oxygen-sensitive subunit, HIF-α, and a constitutively expressed subunit, HIF-β. Under hypoxia, HIF-1α translocates into the nucleus, where it forms a dimer with HIF-1β. This complex then binds to the hypoxia-responsive elements (HREs) of its target genes to restore oxygen homeostasis [77]. 

HIF1 is also involved in glucose metabolism in cancer and increases glycolysis by inducing transcription of the glucose-metabolism-associated genes, including GLUT, LDHA and monocarboxylate transporter 4 (MCT4), or inhibiting mitochondrial function [7]. CircRNAs upregulate HIF1 signaling to elicit the glycolytic phenotype in GI cancer cells. For example, circ_03955 in PC [78], as well as circDNMT1 and circNRIP1 in GC [79,80], upregulate HIF1 expression and promote glycolysis by sponging miRNA-3662, miR-576-3p, and miRNA-138-5p, respectively. 

Conversely, circRNA expression can also be regulated by HIF1 in GI cancers. HIF1 binds directly to *ZNF91* gene promoter to induce the expression of circ-ZNF91 [81]. HIF1 also activates circ-MAT2B transcription through its interaction with the *MAT2B* gene promoter. Subsequently, upregulated circ-MAT2B has been shown to sponge miR-515-5p and increase HIF1 expression, forming a positive feedback loop, which dramatically enhances cell proliferation and glycolysis in GC [82]. 

#### 6.1.2. CircRNAs and C-Myc in GI Cancers

The *Myc* gene family comprises *MYCN*, *MYCL*, and *c-Myc*. As one of the most common oncogenes in eukaryotes, *c-Myc* is ubiquitously expressed in human tissues, and is often activated in various cancers. Upon activation, c-Myc forms a dimer with Max, which binds to the 5′-CACGTG-3′ sequences in the promoter of its target genes to regulate cancer progression [83]. c-Myc also induces the glycolytic phenotype by driving the expression of GLUT1, LDH, HK2, and enolase 1 (ENO1) [84].

Given the vital oncogenic role in human cancers, c-Myc expression is tightly regulated. Wnt/β-catenin signaling induces the translocation of β-catenin to the nucleus, where it is involved in the transactivation of c-Myc [85]. CircSLIT2 serves as a sponge for miR-510-5p to increase the expression of c-Myc, which binds to the LDHA promoter region to upregulate LDHA expression and increase glycolysis in PC [76] (Figure 3). 

#### 6.1.3. CircRNAs and STAT3 in GI Cancers

As a member of the signal transducer and activator of transcription (STAT) family, STAT3 is a highly conserved transcription factor. STAT3 is activated by interleukin 6 (IL-6)-induced phosphorylation. The activated STAT3 then translocates to the nucleus and interacts with the promotor of its target genes to regulate transcription [86,87].

STAT3 is highly expressed in many human cancers. In the TME, STAT3 is constitutively activated to induce cancer progression and immunosuppression [88]. In GC, STAT3 targets the HK2 promoter and upregulates *HK2* gene expression to promote glycolysis. The STAT3 expression is regulated by circRNAs, including circCUL3 [69] and circUBE2Q2 [89], which sponge miR-515-5p and miR-370-3p, respectively, to increase STAT3 expression in GC.

### 6.2. CircRNAs Regulate Glycolysis via the PI3K/Akt/mTOR or FOXK1 Pathways in GI Cancers

A schematic diagram of the mechanism by which circRNAs regulate glycolysis via the PI3K/Akt/mTOR or FOXK1 pathways is shown in Figure 4.

#### 6.2.1. CircRNAs and PI3K/Akt/mTOR in GI Cancers

The PI3K-Akt signaling pathway is involved in the regulation of several cell processes, such as cell cycle progression, apoptosis, and glycolysis [90]. Phosphatidylinositol 3-kinase (PI3K) is a lipid kinase and consists of one catalytic domain (p110) and one regulatory domain (p85). PI3K can be activated by various growth factors (GFs), hormones, and cytokines [91]. GFs activate membrane-bound protein growth factor receptor (GFR), which contributes to PI3K-mediated phosphorylation of phosphatidy-linositol-3,4-bisphosphate (PIP2), forming phosphatidylinositol-3,4,5-bisphosphate (PIP3). Subsequently, PIP3 recruits phosphoinositide-dependent kinase (PDK) and protein kinase B (PKB, also known as Akt) to induce PDK-mediated Akt phosphorylation [92]. 

Mammalian target of rapamycin (mTOR) is a prominent downstream target of Akt. Activated Akt phosphorylates tuberous sclerosis complex 2 (TSC2) and relieves its inhibitory effect on mTOR, leading to mTOR activation and increased synthesis of multiple oncogenic proteins to promote cancer progression [93]. 

CircRNAs are involved in glycolysis through the PI3K-Akt-mTOR axis in GI cancers. CircMYOF increases PI3K expression and subsequently activates Akt to induce glycolysis in PC [94]. CircNRIP1 acts as a sponge for miR-149-5p to upregulate its target, Akt. Increased Akt expression then increases mTOR activity and glycolysis in GC [41]. CircC16orf62 also promotes glycolysis in HCC by activating the AKT/mTOR pathway [95]. 

#### 6.2.2. CircRNAs and FOXK1 in GI Cancers

The Forkhead box (FOX) family is an evolutionarily conserved transcription factor family, which comprises 19 members from FoxA to FoxS. The FOX family contains two domains: a forkhead winged helix–turn–helix DNA binding domain, which directly binds to the promoter region of DNA to regulate gene transcription, and a forkhead-associated (FHA) domain, which interacts with proteins that recognize phosphopeptides [96]. 

FOXK1 is highly expressed in multiple human cancers. Insulin phosphorylates and activates FOXK1 to increase the nuclear translocation of FOXK1 and upregulate its target genes via the PI3K-Akt axis [97]. CircAPLP2 increases FOXK1 expression in CRC by sponging miR-485-5p [98]. 

In HCC, FOXK1 is involved in the glycolytic phenotype by upregulating the transcription of HK2, GLUT1, and LDHA. FOXK1 is a target of miR-1294 and miR-186-5p, both of which are sponged by circ-PRKCI. Thus, FOXK1 expression may be upregulated by circ-PRKCI through miRNA sponging, a process that contributes to enhanced glycolysis in HCC [99].

## 7. Clinical Significance of Glycolysis-Associated CircRNAs in GI Cancers

### 7.1. Diagnostic and Prognostic Value of Glycolysis-Associated CircRNAs in GI Cancers

Although cancer patients diagnosed at an early stage can be cured by surgery, most GI cancer patients are diagnosed at a late stage, leading to a high mortality rate. Therefore, new and effective clinical predictive biomarkers are urgently required to facilitate the diagnosis of GI cancers in the early stage, when treatment may be more effective, but also function as indicators of prognosis, which is important for guiding early intervention in patients with poor prognosis and prolonging their lifespan. 

Glycolysis-associated circRNAs bear important potential as novel diagnostic biomarkers of GI cancers. Due to their covalently closed structure, circRNAs are resistant to exonucleases. Thus, circRNAs are characteristically stable and have longer half-lives than their precursor mRNAs and other noncoding RNAs (microRNAs and long noncoding RNAs). Some glycolysis-associated circRNAs, which are abnormally expressed in the plasma exosomes of patients with GI cancers, have the potential to be used as liquid biopsy biomarkers for early detection of GI cancers [100]. For example, circPDK1 is significantly upregulated in the plasma exosomes and tumor tissues of patients with PC and promotes cell proliferation and glycolysis in vitro and in vivo. As such, circPDK1 may have important diagnostic value in distinguishing patients with PC from healthy controls using this noninvasive technique [101]. 

Accumulating evidence shows that the expression level of many glycolysis-associated circRNAs is associated with the clinical pathological characteristics and survival parameters of patients with GI cancers. Therefore, these circRNAs may have the potential to predict the prognosis of patients with GI cancers. CircCUL3, which is derived from exon 2–3 of CUL3, was found to be expressed at high levels in GC tissues compared with control tissues. This increase in circCUL3 expression is associated with advanced T stage, larger tumor size, and shorter survival time of patients with GC [69]. CircGOT1 was shown to be expressed at high levels in EC tissues and cells. Kaplan-Meier survival curve analysis revealed that high circGOT1 expression is associated with a shorter survival time in patients with EC [53]. In addition, circRPN2 is expressed at lower levels in metastatic HCC tissues compared with non-metastatic HCC tissues. Increased circRPN2 expression inhibits HCC metastasis and glycolysis by regulating the miR-183-5p/FOXO1 axis and increasing ENO1 degradation. Thus, crcRPN2 is implicated as a potential predictor of metastasis in patients of HCC [59]. 

### 7.2. Therapeutic Value of Glycolysis-Associated CircRNAs in GI Cancers

The glycolysis-associated circRNAs are associated with GI cancer progression, metastasis and chemotherapy resistance by regulating cancer-associated signaling pathways and the expression of specific glycolysis-associated enzymes. Thus, these circRNAs have the potential to serve as effective therapeutic targets for GI cancers.

Many studies have shown that circRNAs contribute to glycolysis and GI cancer progression, and silencing of these circRNAs causes the opposite effects in patients with GI cancers. CircSLIT2 was reported to act as a miR-510-5p sponge and upregulate the c-Myc/LDHA axis to promote glycolysis and PC progression. A short hairpin RNA (shRNA) that accurately targets the unique junction of circSLIT2 effectively decreased circSLIT2 expression, thereby inhibiting glycolysis and tumor growth in a mice xenograft model [76]. Similarly, circBFAR is highly expressed in GC tissues and cells and promotes cell proliferation through induction glycolysis. Furthermore, subcutaneous injection of a small interfering RNA (siRNA) targeting circBFAR significantly inhibited tumor growth in vivo [65]. Antisense oligonucleotides (ASOs) are short, synthetic nucleic acid strands that attach to RNA sequences and inhibit their activity [102]. Although ASOs have not been applied to glycolysis-associated circRNAs in GI cancers, ASO-mediated degradation of circRNAs is a promising strategy for cancer therapy. For example, circIPO11, which is highly expressed in HCC tissues, is required for the maintenance of stem cell self-renewal and HCC development. ASOs against circIPO11 were used to decrease circIPO11 expression and inhibit HCC primary tumor progression in vitro [103]. In addition, ASOs targeting circHIPK3 in GC cells were shown to lower circHIPK3 expression and decrease the proliferation and migration [104,105,106]. Overall, inhibition of these oncogenic circRNAs using shRNA, siRNA or ASO represents a promising potential anti-tumor strategy. 

In contrast, several circRNAs have been shown to inhibit glycolysis and negatively regulate GI cancer progression. Therefore, increasing the expression of these suppressor circRNAs represents a novel approach to the therapy of GI cancers. For example, circTADA2A is expressed at low levels in CRC tissues and cells compared with controls. This decrease in circTADA2A expression is associated with larger tumor size, advanced T stage, and distant metastasis in patients with CRC. Exogenous circTADA2A delivered by a vector containing specific DNA cassettes inhibits glycolysis and the cell cycle, and accelerates CRC cell apoptosis. Moreover, in a xenograft model, tumors derived from CRC cells transfected with lentiviruses expressing the circTADA2A sequence exhibited slower growth and reduced tumor volume and weight compared with the mock controls [107]. CircFAM120B, which is reported to be downregulated in CRC tissues and cells, enhances cell proliferation, migration, invasion, and glycolysis. Lentivirus-mediated overexpression of circFAM120B also decreased CRC tumor growth in the xenograft tumor model [108]. Furthermore, the specific lentivirus-mediated enforced expression of circRPN2, circ_0004913, and circ_0001445 in HCC [59,109,110] and circular_0086414 in EC [111] restrained tumorigenesis in vitro.

Similar to the lentiviruses-mediated overexpression of circRNAs, suppressor circRNAs can also be transferred by exosome-based delivery systems. Exosomes, averaging 100 nanometers in diameter, are the most widely studied type of extracellular vesicles (EVs). Exosomes, which are secreted by most eukaryotic cells and surrounded by a lipid bilayer, are ubiquitously present in body fluids, including serum, plasma, urine, and lymph. Exosomes contain proteins, noncoding RNAs, mRNA, and DNA, which are all involved in intercellular communication and play a vital role in cancer progression [112]. Many glycolysis-associated circRNAs, including circFBLIM1 and circ-ZNF652 in HCC [113,114], and circPDK1 in PC [101], are found in plasma exosomes.

Exosomal circZNF91 enters PC cells to sponge miR-23b-3p and increase Sirtuin1 (SIRT1) expression, resulting in increased glycolysis and gemcitabine (GEM) resistance of PC cells [81]. In addition, due to their bilayer membrane, exosomes have a long half-life and can also protect their contents from degradation during the delivery [115]. Furthermore, exosomes function as critical mediators of cell communication by delivering their biologically active contents from one cell to another [116]. Compared with oxaliplatin-sensitive patients, Wang et al. reported that hsa_circ_0005963 is expressed at high levels in the serum exosomes of oxaliplatin-resistant patients, which promotes glycolysis and oxaliplatin resistance in CRC. Mechanistically, exosomes derived from oxaliplatin-resistant CRC cells bind the chemosensitive cells to deliver hsa_circ_0005963 and promote drug resistance by sponging miR-122 and upregulating PKM2 expression [49]. However, improved methods for exosome isolation, introduction of circRNAs to exosomes, and cell-specific targeting of exosomes are urgently required to translate the use of exosome-based delivery systems into the clinic.

Chemotherapy is one of the main treatment options for patients with GI cancers; however, chemotherapy resistance often leads to poor outcomes. Accumulating evidence shows that glycolysis-associated circRNAs play important roles in the sensitivity of GI cancers to chemotherapy. Oxaliplatin is extensively used to treat GI cancers by inhibiting DNA synthesis [117,118]. CiRS-122 is upregulated in oxaliplatin-resistant cells compared with oxaliplatin-sensitive cells, and increases glycolysis and drug resistance of CRC via the miR-122/PKM2 axis [49]. In addition, circ-CCS promotes glycolysis and induces oxaliplatin resistance in CRC [119]. Circ_0094343 has been reported to be markedly downregulated in chemotherapy-resistant CRC tissues, and overexpression of circ_0094343 reduces cell proliferation and glycolysis in addition to improving the oxaliplatin-sensitivity of CRC cells by regulating the miR-766-5p/TRIM67 axis [120]. As another common anti-cancer drug, 5-FU is incorporated into DNA or RNA due to its similarity with the pyrimidines that are fundamental components of these molecules, leading to cell death and inhibiting cancer progression [121]. Increased expression of circSAMD4A in CRC [122] and circNRIP1 in GC [79] contribute to 5-FU resistance by upregulating glycolysis. Therefore, targeting these circRNAs may overcome this type of drug resistance in GI cancers.

## 8. Conclusions

In this review, we have summarized the current body of evidence that glycolysis is increased in GI cancers and contributes to cancer progression. We have also discussed the roles of circRNAs in controlling glucose metabolism by regulating the expression of specific glycolysis-associated enzymes and transporters, as well as cancer-associated signaling pathways. Elucidation of the molecular mechanisms by which circRNAs regulate glucose metabolism during carcinogenesis is critical for an improved understanding of the altered glucose metabolism in GI cancers. With the rapid development of bioinformatics analysis and biotechnology, glycolysis-associated circRNAs offer the potential for use in the diagnosis, prognosis and treatment of GI cancers, thus greatly improving the outcome of patients with GI cancers. 

However, our understanding of glycolysis-associated circRNAs is still incomplete, thus limiting their application in clinical practice. This goal requires the clarification of several issues in future investigations. First, the precise molecular mechanism by which circRNAs are related to glucose metabolism in GI cancers remains to be fully elucidated. For instance, it has been reported that the downregulated expression of circCDC6 in CRC tissues and cells is associated with advanced stage and poor overall survival. Zhao et al. indicated that circCDC6 reprogram glycolysis and CRC progression by sponging miR-3187-3p and upregulating PRKAA2 expression. However, the underlying mechanism by which PRKAA2 regulates glycolysis and tumor growth has not been explored [123]. In addition, the majority of glycolysis-associated circRNAs regulate cancer progression by sponging miRNAs or proteins, although the function of these circRNAs in protein translation has not been reported in GI cancers. Second, although a few abnormally expressed glycolysis-associated circRNAs have been identified in plasma exosomes, most have been detected in GI cancer tissues and cells, which can only be evaluated in tissue biopsies and thus restricts their application as biomarkers in clinical practice. Therefore, further studies are required to identify circRNA expression in samples, such as blood, saliva, and urine that can be obtained via noninvasive techniques.

## Figures and Tables

**Figure 1 cancers-15-02229-f001:**
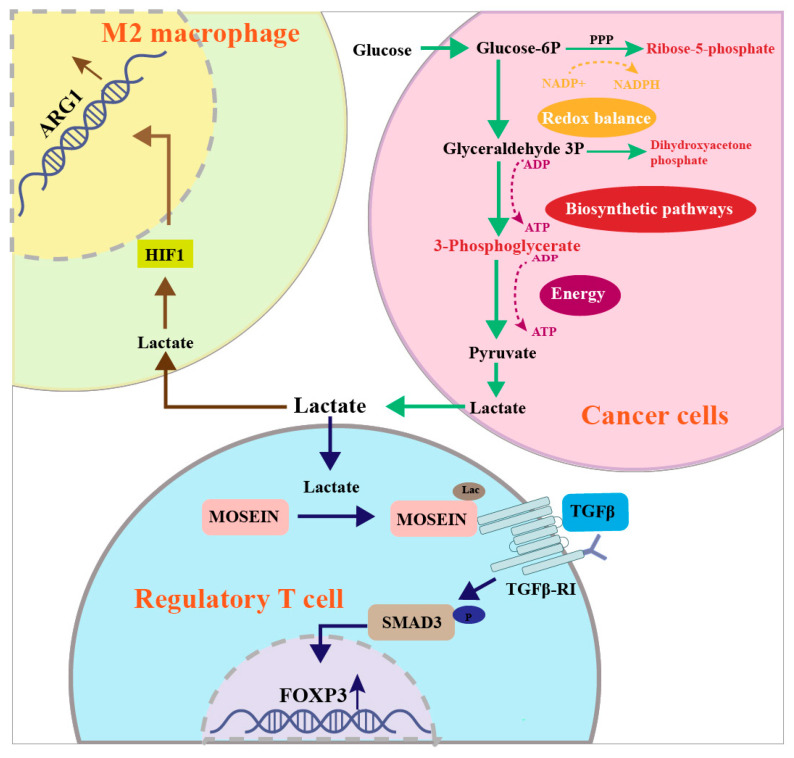
Increased glycolysis contributes to GI cancer progression. Increased glycolysis produces large amounts of ATP and NADH, as well as the raw materials for the biosynthesis of macromolecules required to support uncontrolled cell proliferation. Increased glycolysis also yields abundant lactate, which mediates immunosuppression by inducing M2 phenotype macrophage differentiation and enhancing the function of regulatory T cells.

**Figure 2 cancers-15-02229-f002:**
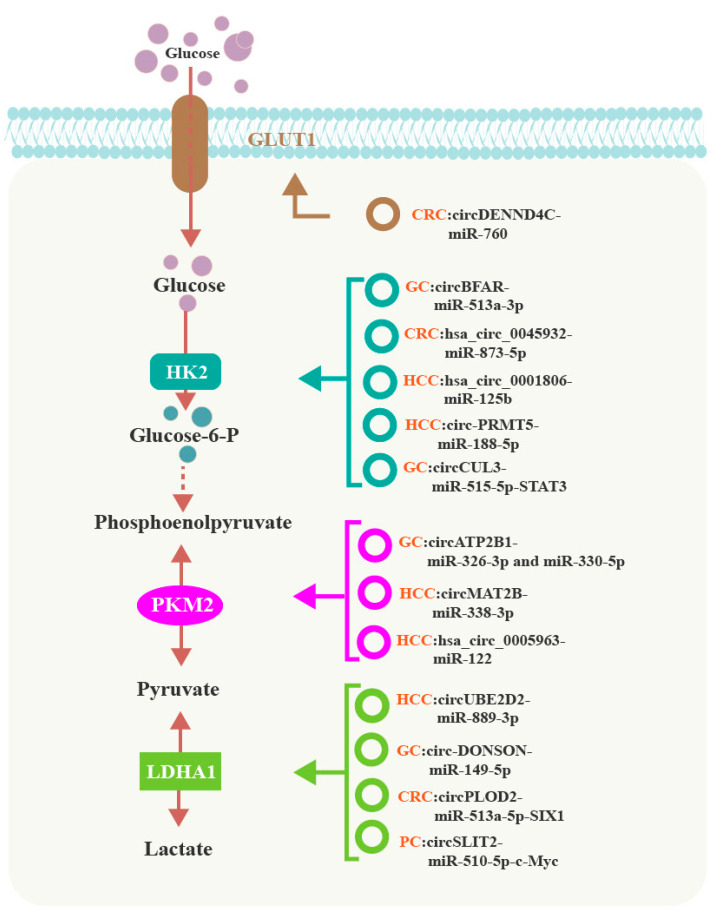
CircRNAs regulate the transcription or expression of GLUT1, HK2, PKM2, and LDHA1 to reprogram glucose metabolism in GI cancers.

**Figure 3 cancers-15-02229-f003:**
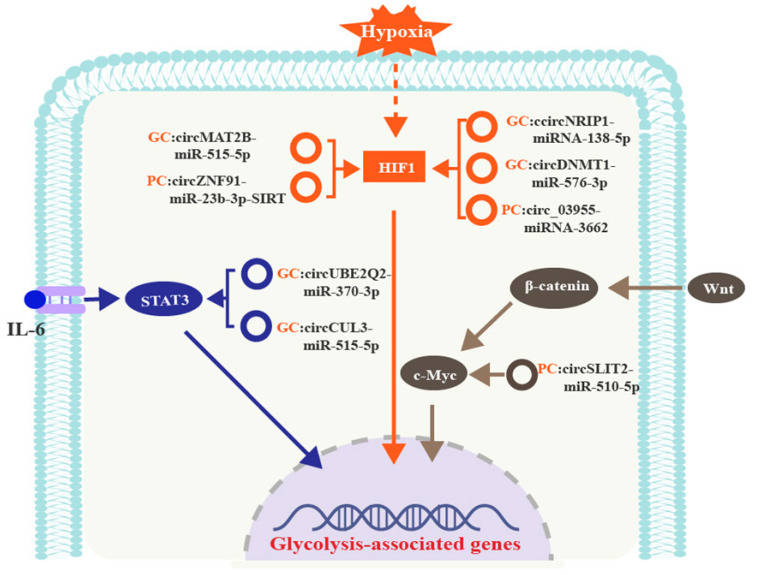
CircRNAs control glycolysis by regulating HIF1, c-Myc, or STAT3 signaling in GI cancers.

**Figure 4 cancers-15-02229-f004:**
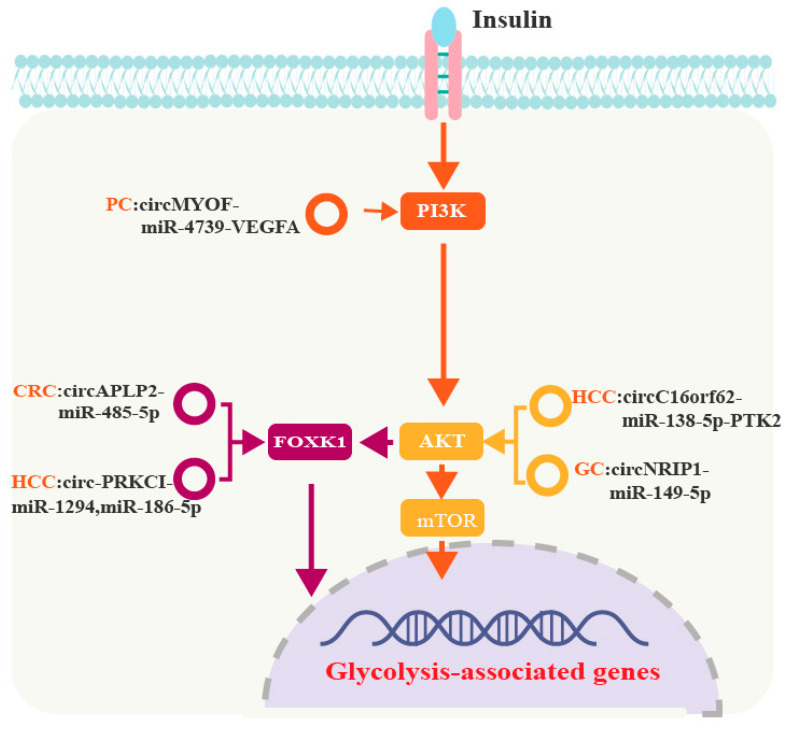
CircRNAs regulate glycolysis via the PI3K/Akt/mTOR and FOXK1 pathways in GI cancers.

**Table 1 cancers-15-02229-t001:** The molecular mechanisms by which circRNAs regulate glucose metabolism in GI cancers.

Regulating	CircRNA	Location	Regulation	Cancers	Involvement of miRNA	Target of Glycolysis	Molecular Mechanism
Transporters and enzymes	circDENND4C	Not investigated	Up	CRC	miR-760	GLUT	Regulates circDENND4C/miR-760/GLUT axis
	circBFAR	Not investigated	Up	GC	miR-513a-3p	HK2	Regulates circBFAR/miR-513a-3p/HK2 axis
	hsa_circ_0001806	Not investigated	Up	HCC	miR-125b		Regulates hsa_circ_0001806/miR-125b/HK2 axis
	circ-PRMT5	Cytoplasm	Up	HCC	miR-188-5p		Regulates circ-PRMT5/miR-188-5p/HK2 axis
	hsa_circ_0045932	Not investigated	Up	CRC	miR-873-5p		Regulates hsa_circ_0045932/miR-873-5p/HK2 axis
	circCUL3	Cytoplasm	Up	GC	miR-515-5p		Regulates circCUL3/miR-515-5p/STAT3/HK2 axis
	hsa_circ_0005963	Exosome	Up	CRC	miR-122	PKM2	Regulates hsa_circ_0005963/miR-122/PKM2 axis
	circATP2B1	Cytoplasm	Up	GC	miR-326-3p miR-330-5p		Regulates circATP2B1/miR-326-3p and miR-330-5p/PKM2 axis
	circMAT2B	Cytoplasm	Up	HCC	miR-338-3p		Regulates circMAT2B/miR-338-3p/PKM2 axis
	circ-DONSON	Not investigated	Up	GC	miR-149-5p	LDHA	Regulates circ-DONSON/miR-149-5p/LDHA axis
	circPLOD2	Cytoplasm	Up	CRC	miR-513a-5p		Regulates circPLOD2/miR-513a-5p/SIX2/LDHA axis
	circSLIT2	Cytoplasm	Up	PC	miR-510-5p		Regulates circSLIT2/miR-510-5p/c-Myc/LDHA axis
	circUBE2D2	Not investigated	Up	HCC	miR-889-3p		Regulates circUBE2D2/miR-889-3p/LDHA axis
Signaling pathways	circPRKCI	Not investigated	Up	HCC	miR-1294 and miR-186-5p	FOXK1	Regulating circPRKCI/miR-1294 and miR-186-5p/FOXK1 axis
	circAPLP2	Not investigated	Up	CRC	miR-485-5p		Regulates circAPLP2/miR-485-5p/FOXK1 axis
	circ-MAT2B	Cytoplasm	Up	GC	miR-515-5p	HIF1	Regulates circ-MAT2B/miR-515-5p/HIF1 axis
	circZNF91	Exosome	Up	PDAC	miR-23b-3p		Regulates circZNF91/miR-23b-3p/SIRT1/HIF1 axis
	circNRIP1	Not investigated	Up	GC	miR-138-5p		Regulates circNRIP1/miR-138-5p/HIF1 axis
	circDNMT1	Cytoplasm	Up	GC	miR-576-3p		Regulates circDNMT1/miR-576-3p/HIF1 axis
	circ_03955	Not investigated	Up	PC	miRNA-3662		Regulates circ_03955/miRNA-3662/HIF1 axis
	circ-PRKCI	Not investigated	Up	HCC	miR-1294 miR-186-5p	FOXK1	Regulates circ-PRKCI/miR-1294 and miR-186-5p/FOXK1 axis
	circAPLP2	Not investigated	Up	CRC	miR-485-5p		Regulates circAPLP2/miR-485-5p/FOXK1 axis
	circSLIT2	Cytoplasm	Up	PC	miR-510-5p	c-Myc	Regulates circSLIT2/miR-510-5p/c-Myc/LDHA axis
	circCUL3	Cytoplasm	Up	GC	miR-515-5p	STAT3	Regulates circCUL3/miR-515-5p/STAT3/HK2 axis
	circUBE2Q2	Cytoplasm	Up	GC	miR-370-3p		Regulates circUBE2Q2/miR-370-3p/STAT3/ HK2 and PFK axis
	circMYOF	Cytoplasm	Up	PC	miR-4739	PI3K/Akt/mTOR	Regulates circMYOF/miR-4739/VEGFA axis which activates PI3K/Akt pathway
	circC16orf62	Cytoplasm	Up	HCC	miR-138-5p		Regulates circC16orf62/miR-138-5p/PTK2 axis which activates the Akt/mTOR pathway
	circNRIP1	Cytoplasm	Up	GC	miR-149-5p		Regulates circNRIP1/miR-149-5p/Akt which activates the activity of mTOR

**Table 2 cancers-15-02229-t002:** The biological functions of glycolysis-associated circRNAs in GI cancers.

Function of CircRNA	CircRNAs	Location	Expression	Cancer Type	Target	Molecular Mechanism
miRNA sponge	circMAT2B	Cytoplasm	Up	HCC	miR-338-3p	Regulates circMAT2B/miR-338-3p/PKM2 axis
	circGOT1	Nucleus	Up	EC	miR-606	Promotes its host gene GOT1 expression by sponging miR-606
	circFAT1	Cytoplasm	Up	CRC	miR-520b	Promotes UHRF1 expression by targeting miR-520b and miR-302c-3p
					miR-302c-3p	
	circATP2B1	Cytoplasm	Up	GC	miR-326-3p	Promotes PKM2 expression by sponging miR-326-3p and miR-330-5p
					miR-330-5p	
Protein sponge	circACC1	Cytoplasm	Not investigated	CRC	AMPK	circACC1 binds the regulatory β and γ subunits of AMPK to increase AMPK holoenzyme stability and activity
	circRPN2	Cytoplasm	Down	HCC	ENO1	circRPN2 binds ENO1 to promote the degradation of ENO1
	circRHBDD1	Cytoplasm	Up	HCC	YTHDF1	circRHBDD1 binds and recruits YTHDF1 to *PIK3R1* mRNA and accelerates the expression of PIK3R1

## Data Availability

Not applicable.

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
