# Peer review of "The Crosstalk and Clinical Implications of CircRNAs and Glucose Metabolism in Gastrointestinal Cancers"

_cancers, 2023, doi:10.3390/cancers15082229_

Round 1

Reviewer 1 Report

This review article that explores how circular RNAs can regulate glucose metabolism in gastrointestinal cancers. Glucose metabolism reprogramming is a hallmark of cancer, and circular RNAs have been found to play a role in regulating it. The article discusses the molecular mechanisms involved and suggests that circular RNAs associated with glucose metabolism could potentially serve as new biomarkers and therapeutic targets for gastrointestinal cancers. Overall, this review highlights the potential importance of circular RNAs in cancer research and the potential for new treatments in the future.

Comments-

1-Recent studies have shown that some circRNAs may function as competitive endogenous RNAs (ceRNAs), meaning that they can bind to and sequester specific microRNAs (miRNAs) that control the expression of target genes involved in glucose metabolism. By binding to these miRNAs, circRNAs may indirectly modulate the expression of these target genes and alter cancer cell glucose metabolism. this need to discussed in detail.

2-Extracellular vesicles (EVs) are microscopic membrane-bound structures that are secreted by cells and contain a variety of molecules, including circular RNAs (circRNAs). EVs may transport circRNAs across cells and alter biological functions such as glucose metabolism in gastrointestinal malignancies. Thus, circRNAs associated with EVs may influence glucose metabolism in gastrointestinal malignancies through both direct and indirect pathways. By obtaining a knowledge of the particular circRNAs and molecular pathways involved, it may be feasible to create novel treatment methods for these forms of cancer; this must also be addressed in the article.

3-The Authors also need to discuss the therapeutic potential of circRNAs in the treatment of gastrointestinal malignancies however which is in its developmental stage. Yet, encouraging studies imply that circRNAs may be used as therapeutic targets or tools in the future. Using antisense oligonucleotides (ASOs) to target certain circRNAs and limit their production in cancer cells is one promising strategy. ASOs are short, synthetic nucleic acid strands that may attach to RNA sequences and inhibit their activity. In a preclinical investigation, researchers employed ASOs to target the circRNA circHIPK3 in gastric cancer cells and discovered that lowering circHIPK3 expression decreased the proliferation and migration of cancer cells.

Author Response

1-Recent studies have shown that some circRNAs may function as competitive endogenous RNAs (ceRNAs), meaning that they can bind to and sequester specific microRNAs (miRNAs) that control the expression of target genes involved in glucose metabolism. By binding to these miRNAs, circRNAs may indirectly modulate the expression of these target genes and alter cancer cell glucose metabolism. this need to discussed in detail.

Thank you for your suggestion. We have discussed the regulatory mechanism by which circRNAs function as miRNA sponges to regulate glucose metabolism in GI cancers in detail. In GI cancers, glycolysis-associated circRNAs function mainly as miRNA sponges. Some circRNAs possess a miRNA response element (MRE), which specifically binds and sponge miRNAs . Glycolysis-associated circRNAs can also regulate their parental genes by acting as miRNA sponges. In addition, circRNAs may be involved in glucose metabolism by binding two or more miRNAs. This information has been added to section 4 of the new manuscript, page 6.

2-Extracellular vesicles (EVs) are microscopic membrane-bound structures that are secreted by cells and contain a variety of molecules, including circular RNAs (circRNAs). EVs may transport circRNAs across cells and alter biological functions such as glucose metabolism in gastrointestinal malignancies. Thus, circRNAs associated with EVs may influence glucose metabolism in gastrointestinal malignancies through both direct and indirect pathways. By obtaining a knowledge of the particular circRNAs and molecular pathways involved, it may be feasible to create novel treatment methods for these forms of cancer; this must also be addressed in the article.

Thank you for your suggestion. As native nanovesicles, exosomes are the most widely studied type of extracellular vesicles (EVs) and have the potential to be used as drug delivery vehicles in cancer therapy. Due to their bilayer membrane, exosomes have a long half-life and protect their contents from degradation during the delivery. Furthermore, exosomes function as critical mediators of cell communication by delivering their biologically active contents from one cell to another. However, improved methods for exosome isolation, introduction of circRNAs into exosomes, and cell-specific targeting of exosomes are urgently required to translate the use of exosome-based delivery systems into the clinic. The relevant content has been added to section 7.2 of the new manuscript, page 13.

3-The Authors also need to discuss the therapeutic potential of circRNAs in the treatment of gastrointestinal malignancies however which is in its developmental stage. Yet, encouraging studies imply that circRNAs may be used as therapeutic targets or tools in the future. Using antisense oligonucleotides (ASOs) to target certain circRNAs and limit their production in cancer cells is one promising strategy. ASOs are short, synthetic nucleic acid strands that may attach to RNA sequences and inhibit their activity. In a preclinical investigation, researchers employed ASOs to target the circRNA circHIPK3 in gastric cancer cells and discovered that lowering circHIPK3 expression decreased the proliferation and migration of cancer cells.

Thank you for your suggestion. Glycolysis-associated circRNAs have the potential to serve as effective therapeutic targets for GI cancers. Many studies have shown that circRNAs contribute to glycolysis and GI cancers progression, and silencing of these circRNAs causes the opposite effects in patients with GI cancers. Antisense oligonucleotides (ASOs) are short, synthetic nucleic acid strands that attach to RNA sequences and inhibit their activity. Although ASOs have not applied to glycolysis-associated circRNAs in GI cancers, ASO-mediated degradation of circRNAs is a promising strategy for cancer therapy. For example, circIPO11, which is highly expressed in HCC tissues, is required for the maintenance of stem cell self-renewal and HCC development. ASOs against circIPO11 were used to decrease circIPO11 expression and inhibit HCC primary tumor progression in vitro. In addition, ASOs targeting circHIPK3 in GC cells were shown to lower circHIPK3 expression and decrease the proliferation and migration. Overall, inhibition of these oncogenic circRNAs using shRNA, siRNA or ASO represents a promising potential anti-tumor strategy. This information has been added to section 7.2 (page 12) of the new manuscript.

Reviewer 2 Report

The article titled "The crosstalk and clinical implications of circRNAs and glucose metabolism in gastrointestinal cancers" provides a comprehensive overview of the molecular mechanisms by which circular RNAs regulate glucose metabolism in gastrointestinal cancers. The authors have conducted a thorough literature search and selection process to identify relevant studies and sources for the review, as the topic is highly specific.

Overall, this review article is very informative, comprehensive, and clear. The authors have done an excellent job of synthesizing a large body of literature on the topic of circRNAs and glucose metabolism in gastrointestinal cancers. The article is well-written and structured, making it easy to understand and follow. However, the only minor drawback is that some of the sections could be slightly more concise, as they contain a lot of information and may be difficult to follow for readers who are not familiar with the topic. Nonetheless, this is a minor criticism, and it does not detract significantly from the overall quality of the article.

Author Response

Thank you for your positive review and constructive suggestions. We have simplified the introduction describing STAT3 signaling (section 6.1.3, page 10) and PI3K/Akt/mTOR signaling (section 6.2.1, page 10-11) to make our review more easy to understand and follow.

Reviewer 3 Report

The manuscript entitled “The crosstalk and clinical implications of circRNAs and glucose metabolism in gastrointestinal cancers” focuses on the discussion of latest advancements in the studying of regulatory roles of circRNAs in metabolic reprogramming in gastrointestinal cancers. The manuscript is well-written and well-organized and can be of interest to the journal audience. However, there some concerns that should be properly addressed before publication.

1.     In the Simple Summary and in the Abstract, the statement “the underlying mechanisms remain unclear” should be modified because (i) this is a contradiction to the next sentence “we discuss the molecular mechanisms…”, (ii) indeed, the mechanisms of circRNA action have been in some extend elucidated.

2.     It is recommended to revise the Abstract because it contains meaningless sentences such as “Multiple mechanisms are related to the regulation of glucose metabolism”, etc.

3.     In the Introduction, the sentence “In the presence of oxygen, pyruvate is then completely oxidized to CO2 and H2O via the tricarboxylic acid (TCA) cycle and oxidative phosphorylation (OXPHOS) in the mitochondria, thereby generating adenosine triphosphate (ATP) as an energy source” should be revised because (i) pyruvate is metabolized through two processes – oxidative decarboxylation followed by TCA; (ii) These two processes produce NADH molecules that further undergo OXPHOS in mitochondrial inner membrane.

4.     In Table 1, make a correction “GULT” for “GLUTs”.

5.     In section 2, in addition to GLUT1 proper discussion of the roles of GLUT3 would be useful. Refs [4,5] should be changed because data on percentage of pyruvate conversion should be supported by original articles, not review articles.

6.     In section 2 and further throughout the text and in Figure 1 legend, there is a very crude mistake – there should be NADH, but not NADPH because NADPH is produced in the pentose-phosphate pathway, not in glycolysis.

7.     Section 3 – a rationale to focus on the immunosuppressive effect of glycolysis should be provided.

8.     Section 6 – it is recommended to locate circRNA characteristics before discussion of their roles in GI cancers, i.e. before section 4. Additionally, section 6 duplicates the in some extend the sections 4 and 5, therefore it is recommended to rearrange the section 6.

9.     English language grammar should be checked and corrected. For example, before Ref. [28], “via” instead of “vis”, etc.

Author Response

  1. In the Simple Summary and in the Abstract, the statement “the underlying mechanisms remain unclear” should be modified because (i) this is a contradiction to the next sentence “we discuss the molecular mechanisms…”, (ii) indeed, the mechanisms of circRNA action have been in some extend elucidated.

Thank you for your suggestion. We have revised the content of the simple summary according to your advice.

  1. It is recommended to revise the Abstract because it contains meaningless sentences such as “Multiple mechanisms are related to the regulation of glucose metabolism”, etc.

Thank you for your advice. The Abstract has been revised accordingly.

  1. In the Introduction, the sentence “In the presence of oxygen, pyruvate is then completely oxidized to CO2 and H2O via the tricarboxylic acid (TCA) cycle and oxidative phosphorylation (OXPHOS) in the mitochondria, thereby generating adenosine triphosphate (ATP) as an energy source” should be revised because (i) pyruvate is metabolized through two processes – oxidative decarboxylation followed by TCA; (ii) These two processes produce NADH molecules that further undergo OXPHOS in mitochondrial inner membrane.

Thank you for your comment. The text has been revised as follows: In the presence of oxygen, pyruvate is metabolized via oxidative de-carboxylation and then enters by the tricarboxylic acid (TCA) cycle. These two processes produce NADH and FADH2 molecules that further undergo oxidative phosphorylation (OXPHOS) in the mitochondrial inner membrane, thereby generating adenosine tri-phosphate (ATP) as an energy source (Section 1, page 2).

  1. In Table 1, make a correction “GULT” for “GLUTs”.

Thank you for your highlighting this error. The manuscript has been corrected accordingly.

  1. In section 2, in addition to GLUT1 proper discussion of the roles of GLUT3 would be useful. Refs [4,5] should be changed because data on percentage of pyruvate conversion should be supported by original articles, not review articles.

Thank you for your comment. We have revised the manuscript as follows: GLUT proteins encoded by SLC2 genes are integral membrane proteins involved in the movement of glucose across the cell membrane. To date, 14 different GLUT isoforms have been identified, each with its own unique tissue distribution. High expression of GLUT1 and GLUT3 has been identified in various cancers and these proteins have been shown to play an important role as glucose transporters in cancer cells (section 5.1, page 8). However, as yet, there have been no reports describing the role of circRNAs in regulating GLUT3 expression in GI cancers.

  1. In section 2 and further throughout the text and in Figure 1 legend, there is a very crude mistake – there should be NADH, but not NADPH because NADPH is produced in the pentose-phosphate pathway, not in glycolysis.

Thank you for highlighting this error. We have corrected the manuscript accordingly.

  1. Section 3 – a rationale to focus on the immunosuppressive effect of glycolysis should be provided.

Thank you for your suggestion. We have revised the manuscript as follows: Initially, the tumor microenvironment (TME) comprises not only cancer cells, but also non-cancerous host cells, including innate and adaptive immune cells. The glycolytic phenotype of cancer cells inhibits the immune defense function of these cells by competitively consuming glucose and restricting the availability of glucose. As the disease progresses, cancer cells generate abundant lactate as the end-product of glycolysis 25. The lactate is then secreted into the TME via the monocarboxylate transporter 1 (MCT1). Although lactate was initially regarded as waste-product of glucose metabolism, accumulating evidence suggests that lactate can induce immunosuppression in GI cancers by modulating the inflammatory microenvironment. Lactate mediates immunosuppression by inducing M2 phenotype macrophage differentiation and enhancing the function of regulatory T cells (section 3, page 5).

  1. Section 6 – it is recommended to locate circRNA characteristics before discussion of their roles in GI cancers, i.e. before section 4.

Additionally, section 6 duplicates the in some extend the sections 4 and 5, therefore it is recommended to rearrange the section 6.

Thank you for highlighting these errors. We have revised the manuscript accordingly.

  1. English language grammar should be checked and corrected. For example, before Ref. [28], “via” instead of “vis”, etc.

Thank you for highlighting these errors. We have revised the manuscript accordingly.

Round 2

Reviewer 1 Report

The Authors have responded to all of my comments and suggestions.